# Assessment of Grain Harvest Moisture Content Using Machine Learning on Smartphone Images for Optimal Harvest Timing

**DOI:** 10.3390/s21175875

**Published:** 2021-08-31

**Authors:** Ming-Der Yang, Yu-Chun Hsu, Wei-Cheng Tseng, Chian-Yu Lu, Chin-Ying Yang, Ming-Hsin Lai, Dong-Hong Wu

**Affiliations:** 1Department of Civil Engineering, Innovation and Development Center of Sustainable Agriculture, National Chung Hsing University, Taichung 40227, Taiwan; mdyang@nchu.edu.tw (M.-D.Y.); g108062401@mail.nchu.edu.tw (W.-C.T.); yiyanlu916@email.nchu.edu.tw (C.-Y.L.); 2Pervasive AI Research (PAIR) Labs, Hsinchu 30010, Taiwan; 3Department of Agronomy, National Chung Hsing University, Taichung 40227, Taiwan; emiyang@nchu.edu.tw; 4Crop Science Division, Taiwan Agricultural Research Institute, Taichung 413008, Taiwan; mhlai@tari.gov.tw (M.-H.L.); dhwu@tari.gov.tw (D.-H.W.)

**Keywords:** machine learning, grain moisture content, smart phone, optimal harvest timing, random forest, support vector regression, feature extraction, smart agriculture

## Abstract

Grain moisture content (GMC) is a key indicator of the appropriate harvest period of rice. Conventional testing is time-consuming and laborious, thus not to be implemented over vast areas and to enable the estimation of future changes for revealing optimal harvesting. Images of single panicles were shot with smartphones and corrected using a spectral–geometric correction board. In total, 86 panicle samples were obtained each time and then dried at 80 °C for 7 days to acquire the wet-basis GMC. In total, 517 valid samples were obtained, in which 80% was randomly used for training and 20% was used for testing to construct the image-based GMC assessment model. In total, 17 GMC surveys from a total of 201 samples were also performed from an area of 1 m^2^ representing on-site GMC, which enabled a multi-day GMC prediction. Eight color indices were selected using principal component analysis for building four machine learning models, including random forest, multilayer perceptron, support vector regression (SVR), and multivariate linear regression. The SVR model with a MAE of 1.23% was the most suitable for GMC of less than 40%. This study provides a real-time and cost-effective non-destructive GMC measurement using smartphones that enables on-farm prediction of harvest dates and facilitates the harvesting scheduling of agricultural machinery.

## 1. Introduction

Grain moisture content (GMC), which represents the maturity of rice [1], is a key indicator for determining the paddy rice harvest period, which could effectively stabilize rice products and further increase farmers’ income. The harvest moisture content (HMC) and rice harvest date determine the green immature grains rate (GIGR), degree of milling whiteness, head rice recovery, bran weight rate, and cracked grains rate, which affect grain quality. In addition, the HMC of diverse varieties affect the quality of different extents [2]. In general, the optimal harvest date (OHD) coincides with a GMC of 25% [3] and head rice recovery is the highest when HMC ranges from 24% to 26% [4]. When grains are mature, the GMC of a quarter of the top of the rice ear is lower but still higher than that of other parts. Considering the quality of dried rice, the OHD occurs when the average GMC reaches 25% [5]. In Taiwan, only HMCs below 32% meet the Standards for the Inspection of Public Stock Paddy and the lower the HMC, the higher the purchase price. Table 1 lists the purchase price of paddy rice for the second crop of 2019, as proposed by the Wufeng Farmers’ Association; for every 1% decrease in HMC, the purchase price increases 2%. High HMC increases the cost and time for drying grains, the damage to grains, and carbon emissions, as well as considerably increases the rate of cracked dried rice, thus decreasing the rice quality [6,7]. In addition, a high HMC is likely to result in a high green immature grain rate. The GMC of the same seed lot can vary considerably; for drying grains with high GMC, grains with a low GMC are overdried until the moisture content is below 14%, causing rates of cracks in the rice, decreasing the quality of the rice, and vastly affecting rice products, revenue, and barn storage. 

Farmers often estimate and judge HMC according to their experience, which is susceptible to psychological factors or the status of neighboring farms. However, growth conditions (e.g., rice variety, time of planting, and farming method) vary among paddy fields; a single appropriate standard for judgment by experience is therefore difficult to establish. Moreover, managers of vast paddy fields, such as agricultural contractors or farmers’ associations, must spend considerable amounts of time and human resources to complete surveys on the growth status of each field. This study employed smartphones to measure the GMC and predict the optimal HMC to help farmers plan the order and time of harvest, which may increase their income and decrease their costs.

Grain maturity initiates with heading: the grain color gradually turns from green to golden yellow. During this period, carbohydrates fill the grains following photosynthesis reactions; this process includes the four stages of milk, soft dough, hard dough, and maturity [8]. During the milk stage, the grains are growing and the inner kernels are filled with milky juice. Grains in the soft dough stage are fuller, although soft. Grains in the hard dough stage turn from green to yellow; they are fully filled and gradually dried, and the average GMC ranges from 25% to 30% as the OHD approaches. At maturity, the grains are mostly filled and dried but late-maturing grains remain in the fill stage; the green on the leaves and clumps gradually fades, the GMC in the main stem is approximately 15–18%, and the average GMC is between 18% and 21%. Rajanna and Andrews (1970) found that in the 8 days following anthesis, GMC gradually increased from 50% to 60%; the proportion of the dry weight of the grains during the fill process was lower and this period was in the milk stage [9]. Subsequently, GMC gradually declined, and the dry weight considerably increased, entering the soft dough stage. Lin et al. (2014) found that grains of the rice variety TNG67 began to swell on the 14th day after heading, with the dry weight accumulating rapidly and entering the soft dough stage; on the 28th day after heading, the grain color turned from green to yellow, entering the hard dough stage [10]. 

During the soft dough stage, GMC varies greatly, ranging from 13% to 43% approximately; the average difference in GMC between the wettest and driest grains was 21% to 29%, with the greatest difference being 46% [4]. As the grains matured, the GMC variation decreased alongside the decrease in the average GMC [11]. Thus, the variation of GMC with high HMC is greater, which leads to uneven drying of grains, increasing the risk of insufficient or excessive drying. The variation of early GMC is greater, affecting the harvested grain’s milling quality. However, in such cases, the environmental risk increases because of the extended waiting period for all the grains on the farmland to be completely filled. Thus, farmers generally harvest during the hard dough stage [12] but determination of the OHD still requires surveys on the actual GMC and consideration of the weather. Thus, repeated on-site measurements of individual GMC is a crucial factor in rice harvesting.

GMC measurements can be either direct or indirect. In direct measurement, moisture in the grains is removed by hot air in an oven. The GMC is obtained by deducting the dried grain weight from the weight of the harvested wet grain; this highly accurate method is the standard for GMC measurement [13]. However, much time is required to complete the test; therefore, the test cannot be implemented immediately and widely to serve as a reference for OHD. 

This study used the direct measurement method to obtain the dry weight. The oven settings of temperature and duration depended on the experience, cultivar, and the state of the crops [13]. The direct measurement method is often used to inspect other methods’ accuracy, such as microwave methods and electronic supplements [14]. The grain pre-process types of the direct measurement method can be divided into broken grains and whole grains by using 2 g grains roasted at 130 °C to obtain the dry weight according to the Association of Official Analytical Chemists’ methods. The cracked grains are prone to evaporating grain moisture, which affects the accuracy of measurement. Therefore, the whole grain method was used in this study, though it took longer.

The GMC range of samples is from 10% to 60%. To ensure the sample stayed in a dry state stably, the sample was placed in an oven until the dry weight value remained constant [15,16]. The high moisture content grains (>50% *w.b.*) were dried at 80 °C [17] in the oven for 7 days until they reached a constant weight. 

Indirect measurements (such as resistance type and capacitance type) are more convenient [5]. Resistance-type GMC-measuring instruments crush grains with a metal board and measure changes in electric currents under certain voltages, which are then converted into GMC. This method relies on resistance as the basis of measurement; when the GMC exceeds a certain level, the resistance inevitably remains fixed. Thus, when the GMC is above 22%, many errors remained despite correction. The system, therefore, has its drawbacks despite advantages such as portability and both simple and rapid operation. With excessively high GMC, the measurement accuracy became inferior. In addition, the destructive sampling procedure used to measure GMC precludes repeated verification and the method might also be affected by temperature and other environmental conditions during measurement [18,19]. Capacitance-type GMC measurement instruments function by placing grains of certain weights and volumes in a high-frequency environment and obtaining the GMC through conversion of the energy absorbed by the hydrogen bonds of water molecules. However, the relationship is non-linear, making it susceptible to variations in temperature and sample density. In general, measurement accuracy is lost with GMC above 25% [20], which suggests this method is unsuitable for grains with GMC above 25%. 

In terms of convenience, smartphone camera functions are highly useful for collecting and using images. In addition to its wide prevalence, relevant data can be rapidly transmitted through the Internet. Related examples include using smartphone images and an external thermal imager for reconstructing three-dimensional thermal images of buildings to determine the buildings’ hotspots or deficiencies on exterior walls [21]; using smartphone images to design an automatic estimation system of calories in food [22]; and using smartphone images to monitor air quality [23]. These studies have highlighted the applicability of smartphones in image analysis. A smartphone functions both as a digital camera and a processor, enabling the user to take images and analyze them. Color analysis of smartphone images can serve as a convenient and portable detector in the field of agriculture. Related applications include the following: simulating the leaf color chart commonly used by rice farmers with images of rice leaves to estimate the required nitrogen fertilizer application rate [24]; using traits of apple diseases to build image recognition systems [25,26]; and estimating the chlorophyll content in the plant according to the colors in the images of paddy rice leaves [27]. The leaf angle can also be quantified using smartphone images [28] to measure fruit size [29]; to estimate apple yield [30]; to automatically measure the leaf area index of crops such as corn, soybeans, and sorghum [31]; to detect the added formaldehyde and residue in milk [32]; to estimate the chlorophyll content and nitrogen content in tea plantations as well as for disease classification; to estimate the chlorophyll content in citrus leaves; to assess the chlorophyll content in corn [33]; to analyze the number of grapevine berries [34]; and to analyze coffee tree branches, estimate its fruit yield, and record information on the geographic locations of coffee trees [35]. Busemeyer [36] published a set of plant phenotyping sensors and proved the feasibility of evaluating plant water content with plant phenotypic spectral information. The PlantTalk in AgriTalk used smart phones to efficiently monitor the growth of plants, such as light-emitting diode lighting, water sprays, and water pumps [37].

Crop appearance is a crucial trait for estimating yield and quality. As an improvement over naked-eye observations, image processing techniques could be applied to the recognition of varieties and surveys on morphological changes. Such techniques could further facilitate the non-destructive evaluation of qualitative traits in terms of the morphological traits, including the size, shape, color, and texture of crops, which may contribute to the determination of the variety, degree of damage, and quality [38]. Yang et al. (2017, 2020) estimated plant height through drone images to assess the rice lodging area and efficiency of variable irrigation management [39,40]. Artificial neural networks were trained to assess 44 formal traits and nine color traits of images in order to discriminate the variety of paddy rice grains; the prediction accuracy of using the three modes (i.e., shape, color, and shape–color) reached 88%, 74%, and 89%, respectively [41]. Magalhães et al. (2021) applied an object detection model to visually detect tomatoes’ ripeness in greenhouses and emphasized that the YOLOv4 Tiny model obtained reliable accuracy by taking only 5 ms of inference time [42] Sanaeifar et al. (2016) used support vector regression (SVR) to establish non-destructive quality inspection of agricultural products according to color traits; the results revealed a significant correlation between color traits and quality [43]. Jimenez-Sierra et al. (2021) used aerial multispectral images through SVR and non-linear autoregressive analysis exogenous model to rice biomass regressors with correlation coefficients of 0.963 and 0.995, respectively [44]. Shinde et al. (2018) used hyperspectral imaging and red, green, and blue (RBG) images to classify naturally ripened bananas and artificially ripened ones [45]; the classification accuracy of the models that were trained using RGB images through random forest (RF) and multilayer perceptron (MLP) reached 98.74% and 89.49%, respectively. To ensure images are suitable for agricultural image analysis, the shooting process should incorporate references on color calibration to avoid errors in image analysis caused by the outdoor environment. The aforementioned studies have revealed the feasibility and advantages of applying smartphone image data analysis to agriculture. In addition, smartphone image analysis enables the non-destructive measurement of GMC; advantages of such techniques, such as correctability and repeated verification, enables users to obtain more information on the grains’ appearance and reduce the cost of measurements. However, limited or no research has been conducted on smartphone-based GMC measurement.

To simplify the data, decrease image noise, and enhance the efficiency of calculation, the most influential elements from the data should be determined before performing the analysis [46]. Yang et al. (2020) used RGB images with color indices’ (CIs) features, such as *ExG*, to effectively enhance the model accuracy of rice lodging identification [47,48]. Five varieties of white, yellow, and mixed corn were discriminated through outward appearance only; the images yielded 28 color indices (YCbCr, HSV, and HLS) through RGB color space transformation; five CIs (RGB computational statistic) were progressively selected according to the mean and standard deviation of the color components to discriminate three varieties of yellow corn [49]. For principal component analysis (PCA), the maximum data variance after vector projection was determined through feature space projection; the principal components with eigenvalues above 1 were retained according to the Kaiser principle [50]; the principal components with the largest explained variances were determined using the screen plot of principal components; and the minimal important factors were used to explain the maximum variance. The plant canopy radius was analyzed using PCA; the number of crown roots and the stem radius better represented the morphological trait of lodging. The effect of seeding rate on lodging-related traits of Gramineae and the yield was also tested [51].

Combined with an on-site survey, this study selected the major CIs by using PCA and compared the four GMC assessment models, including RF, MLP, SVR, and MLR. Predicting GMC using smartphone images served as a non-destructive and convenient approach for GMC measurement that can be applied to large areas and in high frequencies for predicting OHD. The objectives of this research include:Designing a wide range GMC measurement process applicable for the duration of grain maturity based on machine learning technology.Practicing the non-destructive GMC measurement process on smartphones to collect real-time, low-cost, and large-area GMC data in the field.Establishing a multi-day GMC prediction model to predict the future GMC variation for scheduling a suitable harvest time.

## 2. Materials and Methods

This study involved the creation of GMC and OHD assessment models using smartphones in combination with a simple spectral–geometric correction board (SSCB) in rice paddy imaging. The panicle samples whose images were taken were dried to determine the GMC of single panicles. Figure 1 presents the research process and scenario. An image-based GMC assessment model was constructed. The analysis procedures, which include three parts, namely site survey, image processing, and data analysis, are illustrated in Figure 2. 

### 2.1. Field Survey

In this study, the Tainung No. 71 (TNG71) variety of the second crop of 2019 was adopted as the experimental subject for determining changes in GMC of the whole rice paddy over time. According to the Taiwan Good Agricultural Practice, the mean growth days of TNG71 is 104. In this field, spiking was completed on the 71st day; thus, sampling was performed between the 79th and 101st day, with intervals of 1 or 2 days for determining the true value of the on-farm GMC. The sampling was conducted randomly and included spot sampling (single panicle sampling) and area sampling (1 m^2^); GMC measurement and smartphone imaging were implemented for both. Smartphone image data were first preprocessed; the preprocessing included lighting correction, background removal, and CIs calculation to construct an image-based GMC dataset. For single panicle sampling, GMC data with a greater range of variation was collected to increase the prediction accuracy of assessment models for diverse GMCs. Area sampling represented the actual on-farm GMC situation at the time. A multiday GMC prediction model was devised to predict future changes in multiday GMC. 

#### 2.1.1. Field Sampling

The sampling site of this research was located in Farm 24 of the Taiwan Agricultural Research Institute (24°01′48.0″ N, 120°41′34.4″ E) in the Wufeng District, Taichung City (central Taiwan), as presented in Figure 3. The surface of the paddy field was 0.5 ha, the rice variety was TNG71, and the farm employed conventional farming practices. Seedlings were planted on the experimental paddy field on 22 July 2019. Eighteen days before planting, a solution was prepared with 25% tebuconazole, diluted at a ratio of 1:2000 [52], in which the seeds were immersed and sterilized for 24 h. During the nursery period, a solution, prepared with 25% etridiazole diluted at a ratio of 1:2000 [53], was applied again for sterilization; during the first 7 days, the compound fertilizer Taifer #43 was applied as the base fertilizer. The seedling density was 18 cm × 30 cm and both 70% niclosamide and 32% butachlor were applied to prevent golden apple snails and weeds after the seedlings were planted. 

Farm management included managing weeds (once), top dressing (twice), and pest control (twice). Weeds control was conducted between 23 July and 20 August using a solution consisting of bentazon diluted at a ratio of 1:200; fertilizers, including 160 kg/ha of NO.1 instant water soluble fertilizer and NO.4 instant water soluble fertilizer, were applied on 22 August and 10 September, respectively. To prevent rice blast, sheath blight, Cnaphalocrocis medinalis, and bacterial leaf blight, drugs including 10% teclotalam, 10% hexaconazole, 25% buprofezin, and 18.2% imidacloprid were applied on 1 September and 1 October.

In this study, the survey, conducted through on-farm random sampling, began 2 weeks after the heading stage (1 October). Surveys employing single panicle sampling were performed six times over 2 weeks, on 14, 17, 19, 21, 23, and 25 October. The sample sizes for the six surveys were 122, 71, 76, 70, 100, and 95 (total = 534). To obtain the GMC changes over time, collections were performed between 7 and 29 October with intervals of 1 to 2 days; 12 samples of 1 m^2^ were collected each time. The GMC of 201 samples were recorded for 17 days. The ground truth GMC of the aforementioned panicle samples and 1-m^2^ samples were obtained through drying.

#### 2.1.2. Image Devices

The smartphone model used in this study was an iPhone 8; its camera parameters are listed in Table 2. During image shooting, the rice panicle was fixed to the center of the SSCB. The dimension of the board for shooting rice panicle images was 30 cm × 10 cm; it was made of black velvet to reduce reflection. Taking pictures under direct sunlight was avoided and quick-response (QR) codes were attached to the four corners for capturing the region of interest (ROI); the sampling date and sample code were also annotated. A color calibration chart (Spyder Checkr 24, Datacolor^®^, Lawrenceville, NJ, USA) was added to the board for shooting rice panicle images; white, black, and gray pixels (Figure 4a) were used to facilitate image contrast correction and gamma correction. After shooting images of the grain (Figure 4b), the researcher packed the rice panicles in a plastic zipper bag (Figure 4c) to avoid moisture loss during the survey period.

During the outdoor imaging, changes in the lighting conditions might lead to differences in the images and affect the grains’ color in the images. To minimize such lighting-related factors, the smartphone camera parameters were fixed along with an SSCB to facilitate the subsequent gamma correction. 

#### 2.1.3. GMC Measurement

In this study, GMC was measured using the direct drying method; the Denver Instrument SI-234 Summit Series Analytical Balance (Figure 5a) was used to measure the weight of the harvested rice panicle sample (including the bag) and the 1 m^2^ sample was weighed using a JADEVER scale (model: LPWN-1530; Figure 5b). The rice panicle was then placed in an oven at 80 °C to dry for 7 days (Figure 5c). The weight of the dried rice panicle (*W_dry_*) was measured. The wet weight (*W_wet_*) was obtained by deducting the bag weight with the dried panicle from the original rice panicle weight; HMC (%) was represented with the wet basis GMC (*w.b.*). The calculation for *w.b.* is as follows: (1)$$w.b.=\frac{W_{wet}-W_{dry}}{W_{wet}}\times 100\%$$

The results of surveys by Rajanna (1970) [8] and Chau and Kunze (1982) [4] revealed that the valid range of the GMC is 13–60%; thus, 17 samples of the present study were excluded. A total of 517 valid samples were obtained. Figure 6 illustrates the corresponding GMC. The color of the rice panicle changed from green to yellow.

### 2.2. Image Processing

The outdoor lighting conditions were highly variable because of the weather conditions. Each image had to be corrected to reduce the effect of lighting conditions. Image processing was conducted in three stages involving color calibration, ROI cropping, and background removal. The color calibration used in this study included three steps: contrast correction, gamma correction, and halation removal. 

#### 2.2.1. Contrast Correction

When the lighting condition changed, gamma correction was conducted according to the color calibration chart to maximize uniformity when comparing diverse images. This study used histogram stretching to adjust the color balance on images with differences caused by different luminosities. White and black color calibration charts were used as the maximum and minimum, respectively, of the image to calibrate the pixel value. $I_{c}$ was calculated as follows:(2)$$I_{c}=255\times \frac{I-I_{bl}}{I_{w}-I_{bl}}$$
where $I_{c}$ is the pixel value after calibration, *I* is the pixel value before calibration (*R*, *G*, and *B*), and $I_{w}$ and $I_{bl}$ are the pixel values of the white calibration chart and black calibration chart, respectively, before calibration. 

#### 2.2.2. Gamma Correction

Image luminosity was corrected through non-linear gamma correction. A gray calibration chart with a reflectance of 18% was used as the standard for the entire image to adjust each pixel value [54]. The process can be represented as follows: (3)$$I_{gm}=255\times {\left(\frac{I_{c}}{255}\right)}^{\gamma }$$
where $I_{gm}$ is the pixel value after calibration and $I_{c}$ is the pixel value after contrast calibration. The gray calibration chart with a reflectance of 18% in the *L*a*b** color space is (50, 0, 0) or (119, 119, 119), represented with (*R*, *G*, *B*). Thus, the value of *γ* should conform to Equation (4). $I_{g}$ is the pixel value of the gray calibration chart: (4)$$I_{g}^{\gamma }=119$$

#### 2.2.3. Halation Removal

Halation might be produced around rice panicles as a result of reflections from sunlight, leading to some excessively high pixel values. Referring to Ishii et al. (2018) [55], pixels with σ below 35 were considered to have excessively high halation; the standard deviation of the *R*, *G*, and *B* of each pixel was calculated as follows:(5)$$\sigma =\sqrt{\frac{1}{3}\left\{{\left(R_{n}-Ave_{n}\right)}^{2}+{\left(G_{n}-Ave_{n}\right)}^{2}+{\left(B_{n}-Ave_{n}\right)}^{2}\right\}}$$

$R_{n}$, $G_{n}$, and $B_{n}$ are the *R*, *G*, and *B* values of the pixel. $Ave_{n}$ is the mean of $R_{n}$, $G_{n}$, and $B_{n}$, and σ is the standard deviation value of $R_{n}$, $G_{n}$, and $B_{n}$. An image after halation removal is depicted on the right of Figure 7. 

#### 2.2.4. ROI Cropping

QR codes were placed on the four corners of the board used for taking rice panicle images to enable the automatic and consistent capture of the ROI (Figure 8). Code was written using Python 3.7 to enable the detection of the QR codes in the images. The upper left corner provided the origin of the coordinates of the cropped image; the length and the width were fixed according to the range of the sample of the entire panicle length; and the dimension of the cropped area was limited to exclude objects other than the sample and environment background. 

#### 2.2.5. Background Removal

Extraction of the grains required the removal of background images beyond the grains. This study first adopted the *ExG* value (150) of ROI images as the threshold and then removed the pixels of the rubber band (Figure 9a). Subsequently, the hue values above 60 (Figure 9b), as well as the paper card in the image, were removed. Next, the image in Figure 9c was converted to grayscale, pixels with grayscale values above 150 were removed, and the shadow produced by the light source was eliminated. Finally, the rice panicle stem was removed through the hue value and the threshold was modified according to grain color, ranging from 25 to 35. The selection and treatment processes are revealed in Figure 9. HSV, *Gray*, and *ExG* [56,57,58] were calculated as follows.
(6)Gray=0.299×R+0.587×G+0.114×B
(7)ExG=2G−R−B
(8)x=max(R,G,B)
(9)y=min(R,G,B)
(10)$$H_{1}=\{\begin{array}{ll} 0{}^{\circ}, & \mathrm{if}\;x=y\\ 60{}^{\circ}\times \frac{G-B}{x-y}+0{}^{\circ}, & \mathrm{if}\;x=R\;\mathrm{and}\;G\geq B\\ 60{}^{\circ}\times \frac{G-B}{x-y}+360{}^{\circ}, & \mathrm{if}\;x=R\;\mathrm{and}\;G<B\\ 60{}^{\circ}\times \frac{B-R}{x-y}+120{}^{\circ}, & \mathrm{if}\;x=G\\ 60{}^{\circ}\times \frac{R-G}{x-y}+240{}^{\circ}, & \mathrm{if}\;x=B \end{array}$$
(11)$$S_{1}=\{\begin{array}{ll} 0, & \mathrm{if}\;x=0\\ \frac{x-y}{x}=1-\frac{y}{x}, & \mathrm{otherwise} \end{array}$$
(12)$$V_{1}=x\;$$

#### 2.2.6. CI Extraction

This study tested 18 CIs, including *R*, *G*, *B*, *H*, *S*, *V*, *H*, *L*, *S*, *L**, *a**, *b**, *Y*, *Cr*, *Cb*, normalized difference index (*NDI*), green index (*GI*), and red–green ratio index (*RGRI*). HSV was represented by *H*_1_, *S*_1_, and *V*_1_, and HLS by *H*_2_, *L*_2_, and *S*_2_ [46]. The CI of each pixel in the sample was calculated and the mean was adopted to represent the CI of the sample. This was followed by correlation analysis between the obtained value and the GMC measured through direct drying. *H*_2_, *L*_2_, *S*_2_, *L**, *a**, *b**, *Y*, *Cr*, *Cb*, *NDI*, *GI*, and *RGRI* were calculated as follows.
(13)$$H_{2}=H_{1}$$
(14)$$L_{2}=\frac{x+y}{2}\;$$
(15)$$S_{2}=\{\begin{array}{ll} 0{}^{\circ}, & \mathrm{if}\;L_{2}=0\;\mathrm{or}\;x=y\\ \frac{x-y}{x+y}=\frac{x-y}{2L_{2}}, & \mathrm{if}\;0<L_{2}<0.5\\ \frac{x-y}{2-\left(x+y\right)}=\frac{x-y}{2-2L_{2}}, & \mathrm{if}\;L_{2}>0.5 \end{array}$$
(16)k=0.008856
(17)X=0.607×R+0.174×G+0.200×B
(18)Y=0.299×R+0.587×G+0.114×B
(19)Z=0.066×G+1.116×B 
(20)$$L^{*}=\{\begin{array}{ll} 116\times Y^{\frac{1}{3}}-16, & \mathrm{if}\;\;Y>k\\ 903.3\times Y & \mathrm{otherwise} \end{array}$$
(21)$$a^{*}=500\times \left(f\left(X\right)-f\left(Y\right)\right)$$
(22)$$b^{*}=200\times \left(f\left(Y\right)-f\left(Z\right)\right)$$
(23)$$\mathrm{with}\;\;f\left(t\right)=\{\begin{array}{ll} t^{\frac{1}{3}} & \mathrm{if}\;t>k\\ 7.787t+0.1379 & \mathrm{if}\;t\leq k \end{array}$$
(24)Cr=(R−Y)×0.713+128  
(25)Cb=(B−Y)×0.564+128 

All CIs were computed using the OpenCV library of programming functions in Python (version 3.7). In addition, *NDI*, *GI*, and *RGRI* were integrated as predictor variables and both green and red were adopted as the parameters of the *NDI*. Considering the green value of the plant pixels was higher than the red value, vegetal and non-vegetal traits were further foregrounded. *GI* and *RGRI* were used to compare the proportions of red waveband reflection and green waveband reflection [59]: (26)$$NDI=\frac{G-R}{G+R}$$
(27)$$GI=\frac{G}{R}$$
(28)$$RGRI=\frac{R}{G}$$

### 2.3. Image-Based GMC Assessment Model

To construct the GMC assessment model, CIs contributing more to GMC assessment were first selected using PCA. These CIs were then connected with the GMC to design the image-based GMC assessment model. To avoid bias of the testing result, all data processing used K-fold (K = 5) to obtain objective evaluation. The analysis method involved RF, MLP, SVR, and MLR. The model assessment indicators were mean absolute percentage error (*MAPE*), mean absolute error (*MAE*), and root mean square error (*RMSE*). 

#### 2.3.1. Principal Component Analysis

PCA was used to extract the most representative indicators [60]. The principal components with an eigenvalue above 1 or whose cumulative explained variance was above 90% were retained, or indicators with greater eigenvectors were selected for the subsequent analysis. This study standardized 18 CIs, calculated the eigenvalue of their relevant matrixes to eliminate the effects among the assessment indicators, simplified many bulky datasets, and decreased the calculated amount and the complexity of the problem. 

#### 2.3.2. Random Forest

RF is a composite learning algorithm based on decision tree classifiers. A combination of mutually independent decision trees was built randomly. When the sample data were introduced into the RF, judgments could be made at individual decision trees; the one selected most frequently was adopted as the predicted value to construct the RF process. The samples of the bootstrap loading procedure of the ntree were delineated from the raw data. An uncut decision (or regression) tree was constructed for each loaded sample. Rather than selecting the optimal predicted value, each node randomly sampled and selected the predicted values, from which the best splitting variable was selected. Through the prediction of the ntree, the final result was obtained by overall judgment (the results supported by the majority or the mean of regression). 

Considering the decision trees were generated randomly, most of them were irrelevant to the prediction of the original observed value. However, the best prediction path could be determined by observing many prediction results and repeated error correction procedures. The advantage of RF concerns the capability of its fundamental algorithm to process categorical data in continuous data (such as GMC in this study). Moreover, the accuracy of RF is high for most types of data. With a high tolerance for noise, it can be used to calculate the similarity of variable data and estimate the importance of predictor variables. A total of 1000 decision trees were used in this study. 

#### 2.3.3. Multiple Layer Perceptron

MLP repeatedly adjusts the weight and minimizes the difference between the actual output and the predicted output through back propagation. MLP consists of an input layer, a decision (or prediction) output layer, and a hidden layer. The hidden layer has any number of nodes of neurons between the input layer and the output layer. MLP has been widely applied as a prediction model because it enables more complex calculations through its computation structure and mode. 

The advantages of MLP center on its ability to obtain models and trends from abundant, complicated data and its superiority in terms of non-linear data compared with other analytical techniques. Figure 10 illustrates the MLP structure of this study. Eight main CIs were adopted as the input and each of the two hidden layers had 30 neurons. The output was GMC, the activation function was sigmoid, the loss functions were the *RMSE* and *MAE*, the optimizer was RMSProp, and the learning rate was η = 0.001. 

#### 2.3.4. Support Vector Regression

SVR is a machine learning method used in classification and regression predictions. A support vector machine (SVM) identifies a high-dimensional feature space from non-linear data; the feature space is highly correlated with the output data. SVM constructs the optimal hyper plane for classification in the feature space. The SVR operates by inputting the insensitive loss function (ε) into an SVM, which is expanded to solve assessment through non-linear regression. By contrast, SVM inputs the map function φ into a non-linear function, maps the originally non-linear data to a high-dimensional feature space to facilitate data classification in the high-dimensional space, and performs predictions in a high-dimensional feature space. Assume that the training samples are (*x*_1_, *y*_1_), … (*x_n_*, *y_n_*); *x_i_* is input vector and *y_i_* is directed to the output value of *x_i_*; the SVR obtained the function of ε (error maximization) from the training samples. 

The parameter combination of kernel, cost, and ε set by the user determines the complexity of the prediction; the kernel determines the feature space including linearity, polynomiality, radial basis function (RBF), and sigmoid, which are used to verify the robustness of the regression model. RBF is widely used in diverse situations, as well as in this study, whereas other kernel functions are more applicable to specific situations. 

#### 2.3.5. Multiple Linear Regression

MLR is a model used to predict single variance through multiple explained variances. On the basis of the linear relation that exists between the explained variance and the predictor variable, the samples of the predictor variable (*y_i_*) are random and independent from one another; the correlation among explained variances is non-significant; the mean of the residual is 0; and a normal distribution is present. The variance *σ* is calculated as follows [61]:(29)$$\sigma^{2}=\frac{\sum_{i=1}^{n}{\left(x_{i}-\bar{x}\right)}^{2}}{n}$$

The coefficient of determination (R^2^) is the ratio of the total *y_i_* variance that could be explained by the prediction model and ranges from 0 to 1; 0 indicates the impossibility of predicting results by any variance and 1 represents the absence of error in the prediction of the explained variance. However, R^2^ could not recognize the explained variances contained in the model. Furthermore, R^2^ increased as the explained variance increased, although these explained variances were not correlated with the predicted variation. This study integrated the extracted CIs in the MLR model as explanatory variables and adopted GMC as the predictor variable to construct prediction models. 

### 2.4. Multiday GMC Prediction Model

To obtain the OHD, a multiday GMC prediction model was devised. The actual *GMC*_0_ was obtained using smartphones and added to the variance *i* days later ($\Delta GMC_{i}$). The *GMC_i_* of the ith day could be predicted as follows:(30)$$GMC_{i}=GMC_{0}+\Delta GMC_{i}.$$

$\Delta GMC_{i}$ could be as assessed in several ways, such as through the decline of the daily average GMC, the decline rate of the daily average GMC, and the GMC modified equation of the growth period. 

### 2.5. Performance Evaluation

This study used the *RMSE*, *MAE*, and *MAPE* to estimate the model performance and their values were calculated as follows.
(31)$$MSE=\sqrt{\frac{1}{n}\overset{n}{\underset{i=1}{\sum }}{\left(\hat{y}-y_{i}\right)}^{2}}$$
(32)$$MAE=\frac{1}{n}\overset{n}{\underset{i=1}{\sum }}\left|\hat{y}-y_{i}\right|$$
(33)$$MAPE=\frac{100\%}{n}\overset{n}{\underset{i=1}{\sum }}\left|\frac{\hat{y}-y_{i}}{y_{i}}\right|$$

The *RMSE* represents the standard deviation of the difference between the predicted value and the observed value, and indicates the capability of prediction. *MAE* is the mean of the absolute value of the difference between all the predicted values and the observed values. The *MAE* averts mutual cancelation of errors and provides an accurate reflection of the magnitude of the actual prediction error. The *MAPE*, in percentage, represents the ratio of the difference between the predicted value and the observed value, avoiding the effect of the magnitude of cardinality.

## 3. Results

### 3.1. Image-Based GMC Dataset

Single panicles were sampled randomly with intervals of 2 to 3 days. In total, 517 valid samples were obtained. The distribution of GMC measured using the drying method is revealed in Figure 11. The mean of the total sample was 29.32%, standard deviation was 6.98%, median was 27.33%, maximum was 57.43%, minimum was 13.86%, kurtosis was 3.13%, and skewness was 1.81%, revealing that the sample set was concentrated in the GMC range between 22% and 32%; the GMC interval also matched that of the purchased rice (Table 1). To obtain the on-farm GMC spatial variation of the experimental paddy field, paddy rice in 12 lots of 1 m^2^ (approximately nine thickets) were sampled each time. 

GMC was obtained using the drying method. A total of 201 samples were collected during the 17-day sampling period (including three invalid samples). The distribution is illustrated in Figure 12. The distribution of the average GMC of the experimental paddy field ranged between 20.06% and 37.67%; the mean of the total sample was 27.15%; standard deviation was 4.29%; median was 26.45%; kurtosis was −0.39%; and skewness was 0.57%. In contrast to the single panicle sampling, the GMC of samples from the area of sampling followed a uniform distribution; however, the majority of samples were concentrated in the GMC interval between 21% and 28%. 

### 3.2. Image-Based GMC Assessment Model

Figure 13 demonstrates the distribution of the GMC and CIs of the panicle samples. A high correlation was observed between some CIs and GMC such as *H*, *a**, *NDI*, *GI*, and *RGRI*; their *R* values were 0.89, 0.86, 0.86, 0.86, and 0.84, respectively. H represented hue; 0 represented red, 60 represented yellow, and 120 represented green. The *H* values of the current samples ranged between 35 and 65, indicating that when GMC decreased, the hue turned from yellow to reddish. The value range of *a** was ± 127, suggesting that the color components ranged from green to red. Negative values represented a greenish color and positive values represented a reddish color. The distribution of *a** in the current samples ranged from −14.66 to 8.58, indicating a range from greenish to reddish, which is consistent with the trend of change in grains’ surface color. In addition, *R* and *G* were adopted as the operation parameters for *NDI*, *GI*, and *RGRI*. A summary of the aforementioned findings indicated that, as the grain maturity increased, GMC changes were more sensitive to CIs, related to green and red wavebands.

To extract the main explanatory variable of the assessment model, PCA was used to extract multivariate traits (Table 3 and Figure 14). The explained variance of PC1, PC2, and PC3 was 44.7%, 30.4%, and 24.3%, respectively. It was demonstrated that the greater the cumulative explained variance, the greater the importance of the principal component. This study selected eight CIs, namely *H*_1_, *a**, *G*, *S*_2_, *L**, *B*, *S*_1_, and *b**, to build the assessment models.

Four GMC assessment models were devised; the distributions of their assessed values and actual values are revealed in Figure 15. Regarding the RF and MLP models, 80% of the 517 samples were adopted as the training data, with the remaining samples serving as the testing data. The same testing samples were adopted for the SVR and MLR models. The R^2^ of the four models, namely RF, MLP, SVR, and MLR, were 0.88, 0.90, 0.91, and 0.91, respectively. The resultant GMC obtained by the four models and the measured values were all highly correlated, and the explained variance of the SVR and MLR were greater. Table 4 presents the model efficacy assessment results. The testing data included all the samples, samples with a GMC below 40%, and samples with a GMC below 32%, with sample sizes of 103, 86, and 77, respectively. Globally, the *RMSE* of the four models was between 2.49% and 2.98%, and the *RMSE* of the MLR model was the lowest (2.49%). Regarding the *MAE* assessment, the lowest value was observed in the MLR model (1.71%). The *MAPE* ranged from 5.24% to 6.28%, with that of the SVR being the most favorable. The model prediction error for samples with a GMC below 32% was less than that for all the samples and for samples with a GMC below 40%. The *RMSE* ranged from 1.53% to 1.80%; the *RMSE* of the SVR model was the lowest and that of RF was the greatest. The *MAE* ranged from 1.08% to 1.41%. The *MAE* of the SVR model was the lowest and that of RF was the greatest. 

### 3.3. Image-Based GMC Prediction Model

To predict the OHD of rice, multiday GMC prediction models were required. The current on-farm $GMC_{0}$ was represented by the average GMC of the 12 samples collected through daily area sampling to predict the future $GMC_{i}$. Multiday $GMC_{i}$ was predicted by calculating the modified equation of the average daily variation, the average daily change rate, and the number of growth days. This study tested the linear model, the indexical model, and the logarithmic model as follows.

The variation of daily GMC is assumed to be a fixed value. The calculated average daily variation is −0.66%, enabling the calculation of the decline of daily GMC. The $\Delta GMC_{i}$ of the ith day is determined as follows: (34)$$\Delta GMC_{i}=-0.66\%\times i.$$

The daily change rate of the GMC is assumed to be a fixed value. The calculated average daily change rate is 0.969%, enabling the calculation of the decline rate of daily GMC to predict $\Delta GMC_{i}$ of the ith day as follows: (35)$$\Delta GMC_{i}=0.969^{i}.$$

In addition, as demonstrated in Figure 16, the growth days and GMC were highly correlated logarithmically, enabling the construction of a logarithmic model (Equation (36)) between them, with R^2^ being 0.85. Following the estimation by the image-based GMC assessment model, GMC did not contain information on the growth days. This model was thus used to calculate the growth days (*GD*) corresponding to the $GMC_{0}$.
(36)$$GD=-40.78\ln\left(GMC_{0}\right)+36.02$$

Equation (37) and Figure 17 demonstrate the logarithmic formula through the regression of the growth days and GMC; R^2^ was 0.84. The formula for predicting the $GMC_{i}$
*i* days later, according to this model, can be represented as follows:(37)$$GMC_{i}=-0.514\ln\left(GD+i\right)+2.58$$

### 3.4. Model Performance

To verify the efficacy of the decline of daily GMC (Equation (34)), decline rate of daily GMC (Equation (35)), and the GD-based GMC prediction model (Equation (37)), the harvest date assessment model predicted the daily GMC of the subsequent 1 to 8 days (1 ≤ i ≤ 8) according to the multiday GMC assessment dataset, followed by a comparison with the actual GMC. The results indicated that the variation in the *MAE* of the decline of daily GMC ranged from 0.67% to 2.23%; the decline rate of daily GMC ranged from 0.56% to 2.08%; and the GD-based GMC prediction model ranged from 0.83% to 1.21%. Despite the minimal error in the decline rate of daily GMC on the second day, the *MAE* trends of the decline of daily GMC and decline rate of daily GMC increased over time, whereas the trend of the GD-based GMC prediction model was more stable, relative to the aforementioned two models, and its average *MAE* over the 8 days was the lowest. 

## 4. Discussion

### 4.1. Performance of the Image-Based GMC Assessment Model

This study used changes in the phenotypic spectrum of grains to assess the GMC and applied the threshold of CIs to the extraction of grains from images of single panicles. Shei and Lin (2012) [62] used the hue (also referred to as HLS) values between 40 and 90 degrees to extract brown rice stems. The current study performed the image calibration procedure and applied a hue value of between 25 and 35 degrees to remove paddy rice stems; however, the range of color change varied among rice grains of different varieties.

In this study, an image-based GMC dataset was constructed using smartphone imaging and on-site sampling of single panicles. The range of GMC distribution was consistent with that of prior research results, indicating that GMC at maturity followed a left-skewed distribution, with the difference between the highest and lowest on-farm GMC being as high as 43.57%. The GMC of the samples in this study were mainly concentrated between 22% and 32%, accounting for 78% of the total sample size, whereas the resultant GMC of single panicle sampling effectively reflected the difference regarding the actual on-farm GMC. 

Linear analysis and PCA on the 18 CIs revealed that *H*_1_ (R^2^ = 0.89) and *a** (R^2^ = 0.88) were the most relevant components in PC1, with the greatest explained variance (44.17%), and could be used to construct GMC assessment models; this result is similar to that of Ishii et al. (2018). Most grains in the soft dough stage are immature; when the rate of green immature grains is high, the GMC variation increases and the grain quality becomes inferior. This is likely to increase the time and cost for drying and decrease the price as well as grain merchants’ willingness to purchase. The current analysis revealed that *a** remained in the greenish range when GMC > 32% and in the reddish range when GMC < 32%. Thus, the current researchers inferred that the growth stage progressed from the soft dough stage to the hard dough stage as the GMC gradually decreased to 32%, entering the GMC range of grains purchase. The OHD prediction also yielded better outcomes.

In a general estimation, the *MAE* ranged between 1.71% and 2.04%. The global differences among the models could be attributed to the following two factors. First, grains with a high GMC varied to a larger extent; grains with a GMC above 32% were in the soft dough stage, leading to the larger variation range of GMC and greater difference. Second, the current sampling time was performed randomly 2 weeks after spiking; the GMC value distribution was skewed to the left. Thus, the number of samples with a high GMC accounted for a lower proportion. 

The GMC sensor, based on the portable resistance, was used for measuring in situ GMC, while the capacitance to sample more than 200 g grains is generally needed. Table 5 shows the comparison of different operation principle GMC sensors. The specifications of the GMC sensors refer to the official product specifications.

The suitable range of GMC sensors is below 20% *w.b.*, which is much less than the harvest moisture content, and the process must be conducted by a destructive approach. The Image-GMC assessment model proposed in this study is applicable to measure the range of between 40% to 20% GMC, which is flexible, universal, and non-destructive, and allows for in situ measuring repeatedly. The model can also assess the current GMC and predict the future trends in advance to realize the optimal harvesting time and proper scheduling of agricultural machinery.

In testing, when GMC was above 20%, the error rate of the GMC measured using resistance grain moisture meters was above 4%. The four prediction models in the current study yielded their optimal performance with GMC below 40%, with the assessment *MAE* between 1.23% and 1.63%; below 32%, with the assessment *MAE* between 1.08% to 1.41%, enabled the prediction of a wider range of GMC. Considering a GMC of 32% indicates the crop is already in the harvestable stage, GMC prediction could be improved if the trend of GMC variation is determined earlier in the actual application. In the future, immature rice panicles with greenish color could be adopted as the training samples to overcome the aforementioned concerns that might affect model prediction errors at high GMCs. 

To verify the applicability of the proposed model, samples of different growing conditions and rice varieties were tested. Another dataset was taken in the first crop season in 2020 with different growth conditions from the training dataset (the second crop season in 2019). The Image-GMC assessment model has been preliminarily confirmed as its practical extension to different growth conditions and varieties of rice (see Table 6). Based on the Image-GMC dataset, the SVR model has the best inference result (*MAE* = 3.31%).

The images of different rice varieties as test samples, which contain early maturity rice (TNG71) and a middle-late maturity rice (Tainan No.11, TN11). The Image-GMC assessment model was established on the basis of the TNG71 dataset to infer the images of TN11. The preliminary result shows that the SVR model had the best performance, with a *MAE* of 3.66%.

Comparing the test results in Table 6, the testing *MAE* was doubled for different growing conditions and different varieties, indicating an improvement of the inference model needed in the future work. Importing common features between training data and unknown data by integrating feature transfer methods or self-adaptive learning methods could strengthen the model applicability. 

Furthermore, rice growth and grain development are affected by the weather and environment. A future study should include meteorological prediction, such as solar radiation, temperature, air pressure, humidity, and wind speed, for better establishing a GMC prediction model with environmental interactions.

### 4.2. Performance of the Moltiday GMC Prediction Model

Considering the substantial GMC variation in single panicles, the main difference between the multiday GMC dataset sampling and the single panicle sampling was that the multiday sampling was performed according to the average grain GMC value per square meter. Compared with the GMC sampling, employing the image-based GMC dataset, the range of distribution of the average GMC per square meter in the multiday GMC dataset was smaller and the variation was less, with the lower kurtosis indicating a more uniform distribution. The multiday GMC dataset was similarly skewed to the left, which could reflect the difference in the GMC distribution of diverse sampling methods (Table 7). Thus, this dataset was used to further train the multiday GMC prediction model for predicting multiday GMC and harvest time. 

The more accurate prediction results for GMC in the subsequent 3 days was related to the declining rate of daily GMC, whereas the modified equation of the growth period of the GMC after 3 days or more was better (Figure 18). The main difference was that the prediction results of the decline of daily average GMC and the decline rate of daily average GMC during the training stage were affected by the accumulation of multiday errors, whereas the GD-based GMC prediction model was not affected by such errors. This study adopted the GMC of the second crop of 2019 as the testing data and verified the applicability of the three models with novel data from actual fields. The *MAE* of the testing data is revealed in Figure 19; the GD-based GMC model prediction yielded favorable results in the subsequent 2 days and after. 

In the GMC model sampled per square meter and based on the growth days, the average GMC revealed a gradual daily decline. The CIs of images shot with smartphones were adopted as estimation indicators for grain HMC. Compared with current grain moisture meters, the present method yielded more favorable estimation results when GMC was below 32%, enabling diverse users such as farmers’ associations or agricultural management units to estimate grain drying cost and rice quality. Combining single-day GMC assessment models and multiday GMC prediction models by employing smartphone images might serve as a valuable reference for OHD-related decision-making.

### 4.3. Futrue Work

In the future, the image segmentation of convolutional neural networks, such as U-Net and SegNet, could be employed for removing background images beyond the grains, which would hopefully decrease the interference of background noise more efficiently. GMC changes over time and is affected by weather factors such as sunlight, wind, and rain. In future studies, historical weather data and short-term weather forecasts can be employed to revise the GMC predictions, followed by chronological analysis. For example, recurrent neural networks might use long short-term memory to predict long-term GMC changes and OHD by using historical data. This could help users manage farm harvest work and contribute to the quality and quantity of harvested grains, in addition to helping farmers track changes in farm production over time.

The leaf color chart produced by the International Rice Research Institute, as the standard of comparison, found that the nitrogen content in plants was highly correlated with paddy rice leaf color indices and could serve as a tool for judging the nitrogen fertilizer application rate. However, a corresponding panicle color board for GMC measurement is not available. Thus, in addition to assessing GMC through smartphone image data, future research could consider a panicle color board as a simple tool for measuring GMC; adopt the average of samples with ± 1% GMC to represent color changes by using H_1_, S_1_, and V_1_ values; and design a panicle color board as shown in Figure 20. 

GMC prediction procedures employing mobile devices could also use App to adapt to large-scale GMC surveys. The Global Navigation Satellite System coordinates of each image could be connected and recorded through cloud servers, and the results of the measurement could be saved in a database. In this manner, the coordinates of the images and the resultant GMC from the prediction analysis can be presented using a geographic information system. To enable farmers to determine the OHD with GMC information, the results can be presented in the form of action proposals with three categories: inappropriate for harvest (GMC ≥ 32%), appropriate for harvest (32% > GMC ≥ 28%), and “the most appropriate for harvest” (GMC < 28%). Such a system would help farmers determine the OHD. Users with numerous contracting farmers, such as large-scale agricultural contractors and farmers’ associations, could rapidly determine the HMC of individual paddy fields, schedule harvester dispatch, and plan harvest priority, which could vastly decrease the cost of grain drying and increase rice quality and farmer income. 

## 5. Conclusions

The difference between the GMC of the wettest and driest grains reached 46% in spot sampling and 18% in area sampling, and the average GMC followed the daily decline, revealing that GMC measurements with single panicles as the unit accurately reflect the difference of actual on-farm GMC. Therefore, the smartphone-based GMC measurement proposed in this study assessed GMC_0_ and was implemented using an image-based GMC assessment model. *GMC_i_*, *i* days later, was predicted using a multiday GMC prediction model, which rendered GMC measurement more rapid and convenient, and effectively increased the sample size. Moreover, the measurement was non-destructive, avoiding misjudgment of the average GMC due to on-farm differences.

This study used images of paddy rice panicles shot under visible light and adopted their *R*, *G*, and *B* values as GMC estimation indicators. Color space transformation revealed that CIs related to red and green wavebands (such as *H* and *a**) were highly correlated with HMC and suitable for GMC assessment. The R, G, and B values of images shot outdoors were susceptible to sunshine variation. Thus, this study produced an SSCB as the basis for gamma correction, which facilitated actual farmland application. Although the assessment accuracy of the models varied for diverse GMC intervals, CIs could accurately reflect the grain GMC when the GMC was below 32%, remedying the limits of portable resistance grain moisture meters in terms of the range of application. Measuring GMC was conducted through machine learning, including random forest, multilayer perceptron, support vector regression, and multivariate linear regression, in which the SVR model with a *MAE* of 1.23% was the most suitable for a GMC of less than 40%. The smartphone imaging method greatly reduced the equipment cost for non-destructive GMC measurement instruments.

This study proposed a smartphone-based GMC measurement tool that obtains and analyzes a considerable number of morphological traits in grain images, which enables its wide application. The objective and contributions achieved in this study are as follows:The GMC assessment model applying the SVR algorithm has a great performance with a *MAE* of 1.23% for a GMC of below 40% and a *MAE* of 1.08% for a GMC of below 32%, so as to perceive the GMC variation in the field at a very early stage.The proposed non-destructive GMC assessment model executed through smartphones is low-cost and handy so as to efficiently collect the field GMC over a broad area in time.The proposed multi-day GMC prediction model provides the prediction of daily GMC variation for the coming week, which helps to evaluate the best harvest timing and optimize the scheduling of agricultural machinery.

## Figures and Tables

**Figure 1 sensors-21-05875-f001:**
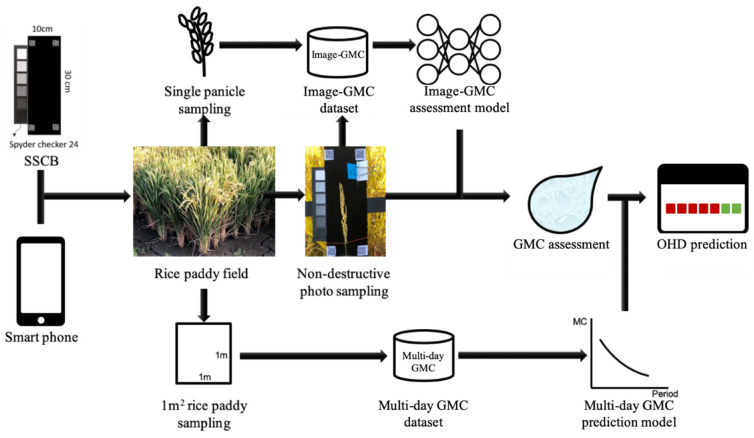
Research process and scenarios.

**Figure 2 sensors-21-05875-f002:**
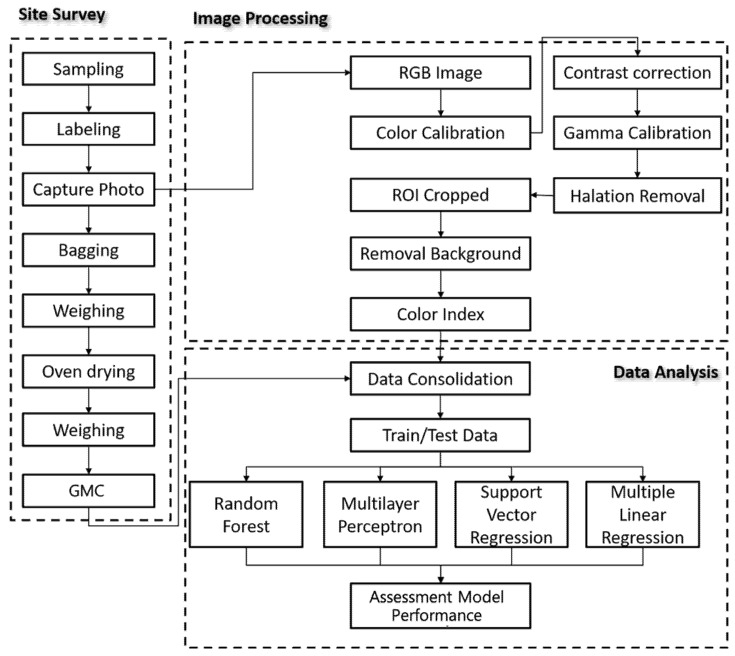
Study Framework.

**Figure 3 sensors-21-05875-f003:**
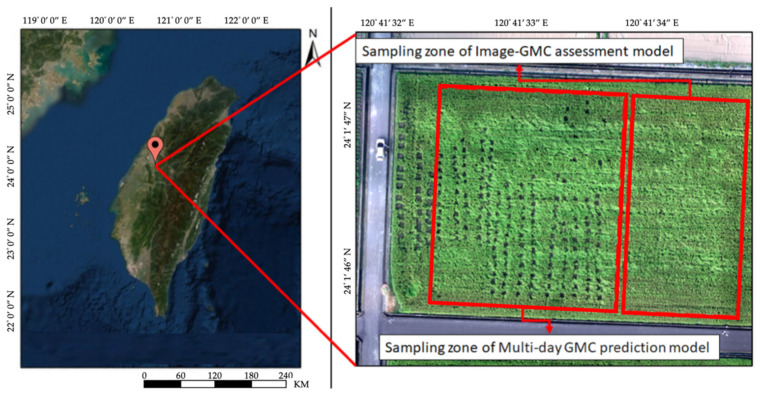
The study site at the Taiwan Agricultural Research Institute.

**Figure 4 sensors-21-05875-f004:**
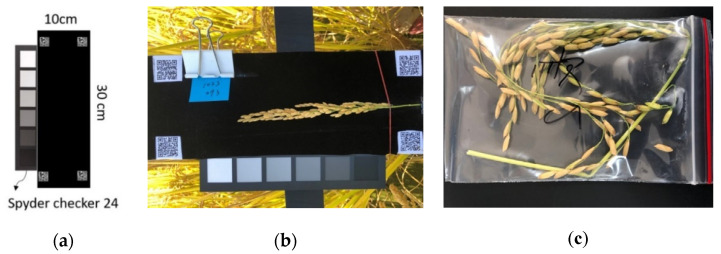
Smart phone images with SSCB (**a**,**b**) and bagged sample (**c**).

**Figure 5 sensors-21-05875-f005:**
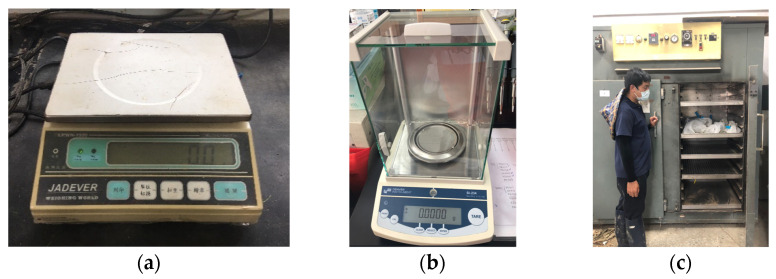
Multi-day GMC dataset grain weight was measured by JADEVER LPWN-1530 (**a**) and image-based GMC dataset grain weight was measured by the Denver Instrument SI-234 Summit Series Analytical Balance (**b**). These samples were all dried in an oven at 80 °C for 7 days (**c**).

**Figure 6 sensors-21-05875-f006:**
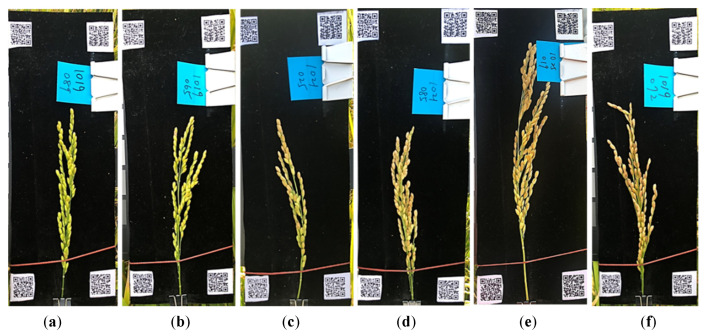
The phenotype transformation of single panicles from green to yellow with moisture content (% *w.b.*): (**a**) 51, (**b**) 44, (**c**) 35, (**d**) 31, (**e**) 29, and (**f**) 24.

**Figure 7 sensors-21-05875-f007:**
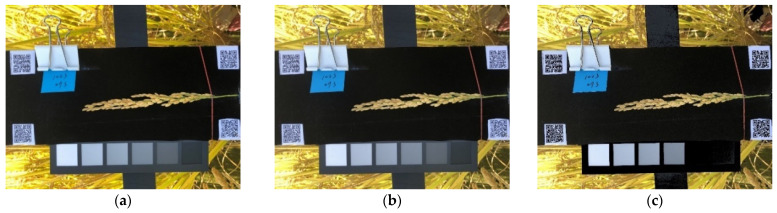
Image color calibration including contrast correction (**a**), gamma correction (**b**), and halation removal (**c**).

**Figure 8 sensors-21-05875-f008:**
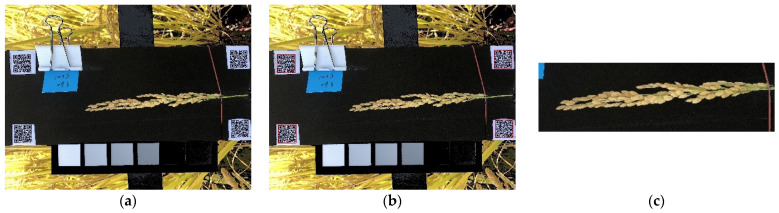
ROI cropped display. Corrected image (**a**), QR code detection (**b**), and ROI area (**c**).

**Figure 9 sensors-21-05875-f009:**
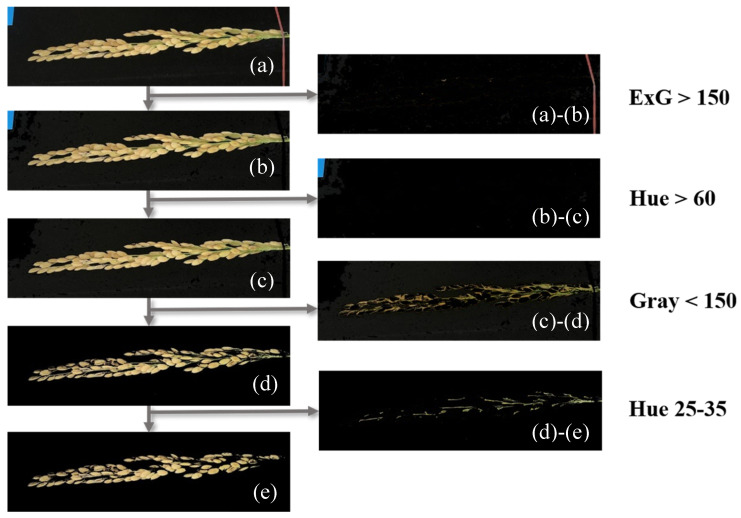
Background removal process included the (**a**) ROI image, (**b**) ROI image with ribbon removal, (**c**) sample card removal from image (**b**), (**d**) shadow removal from image (**c**), and (**e**) stem removal from image (**d**).

**Figure 10 sensors-21-05875-f010:**
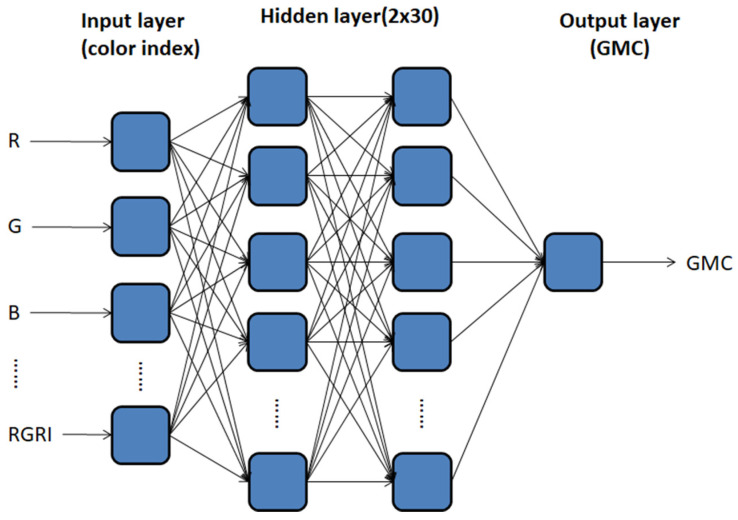
MLP structure of Image-GMC assessment model.

**Figure 11 sensors-21-05875-f011:**
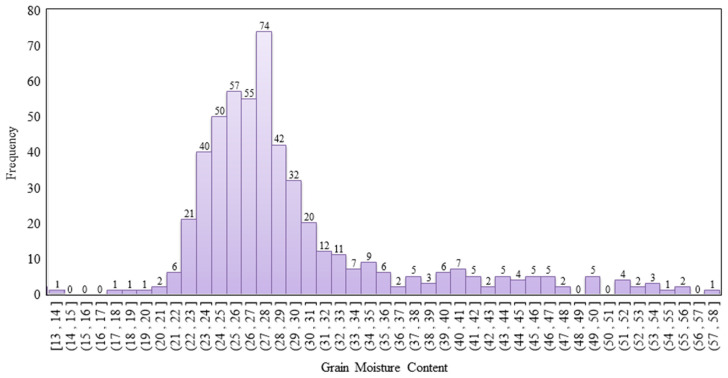
The distribution of the GMC among single panicle samples (N = 517, [ ]: closed interval, and ( ): open interval).

**Figure 12 sensors-21-05875-f012:**
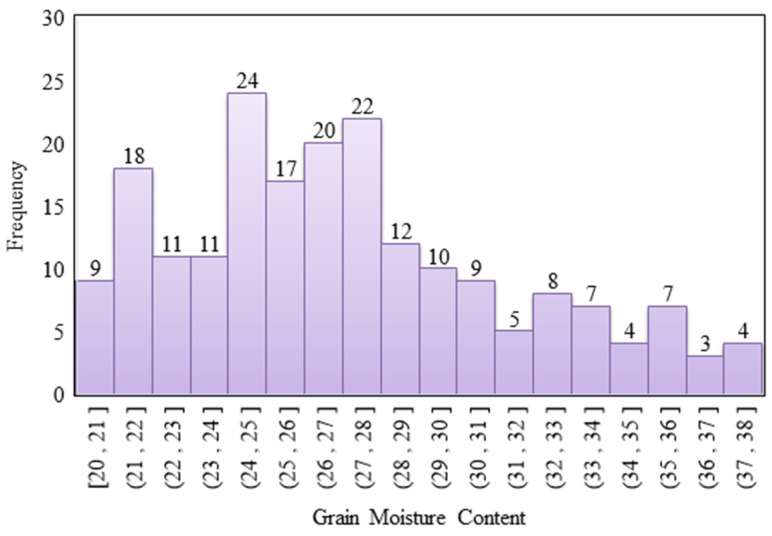
GMC distribution of the decline mode (N = 201).

**Figure 13 sensors-21-05875-f013:**
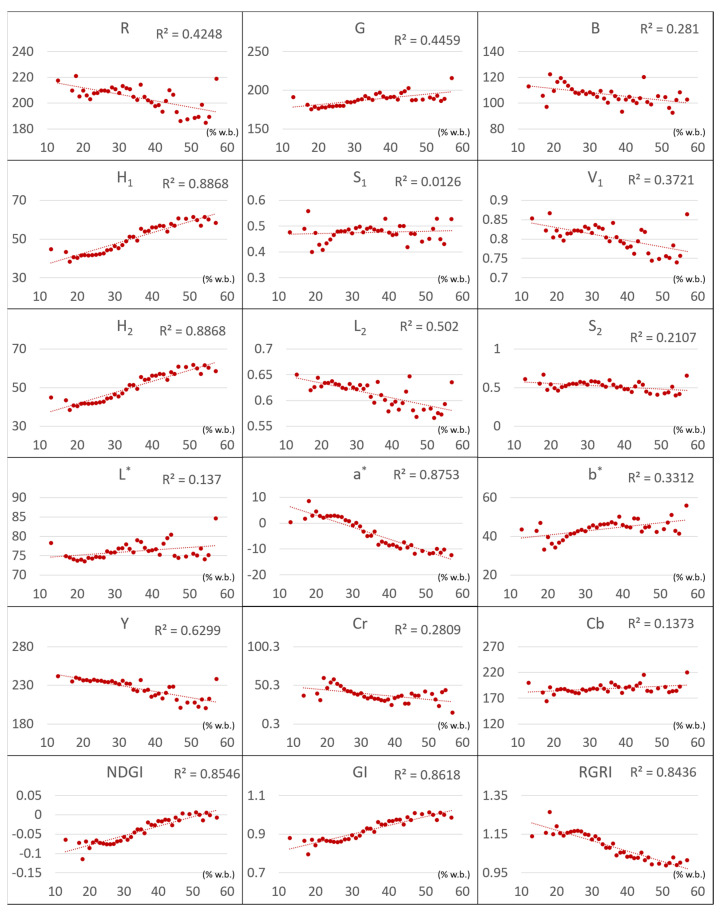
The distribution of color index and different GMC (x-axis: % *w.b.* and y-axis: value of CIs).

**Figure 14 sensors-21-05875-f014:**
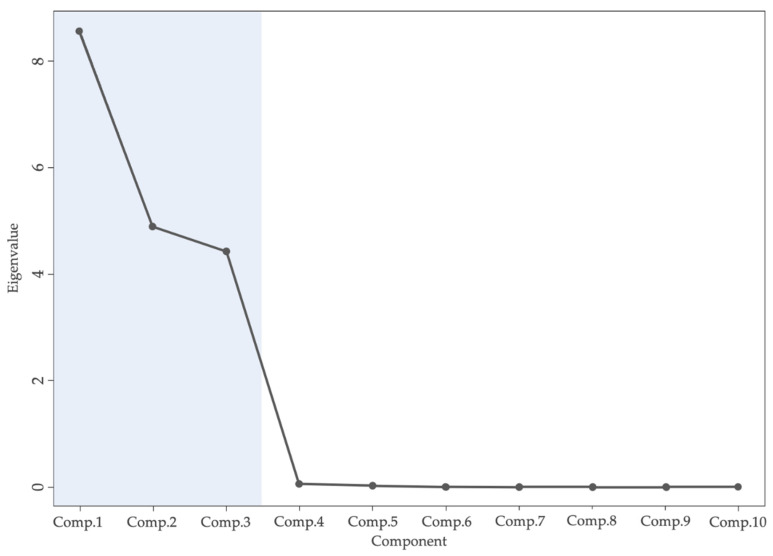
Screen plot of principal components.

**Figure 15 sensors-21-05875-f015:**
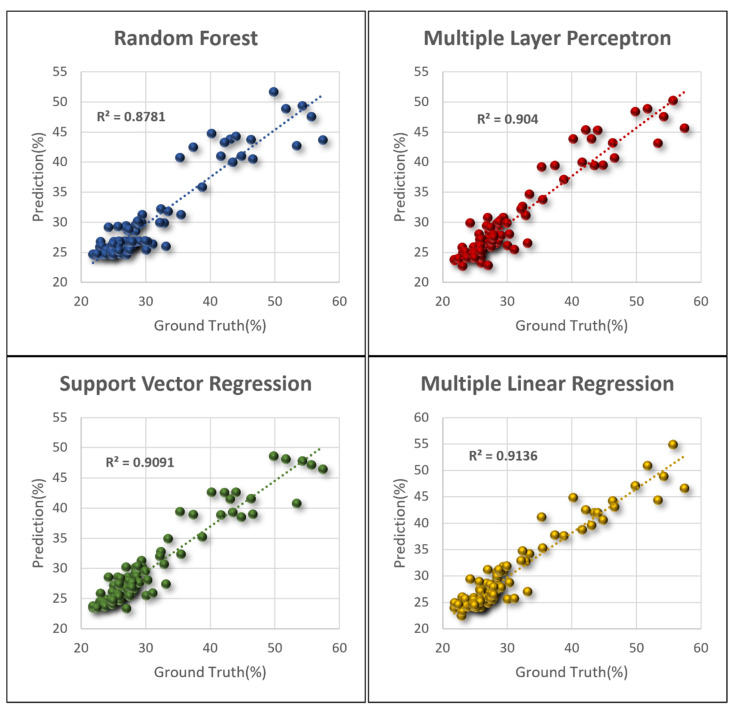
GMC distribution of prediction versus actual data.

**Figure 16 sensors-21-05875-f016:**
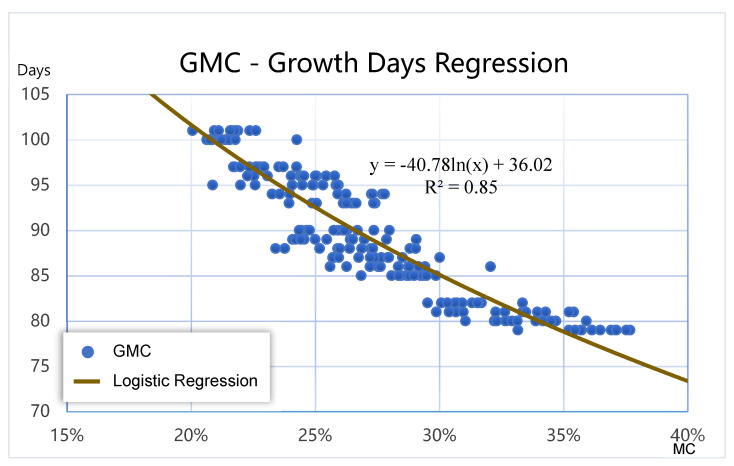
Data distribution and logistic regression of the GMC and growth days.

**Figure 17 sensors-21-05875-f017:**
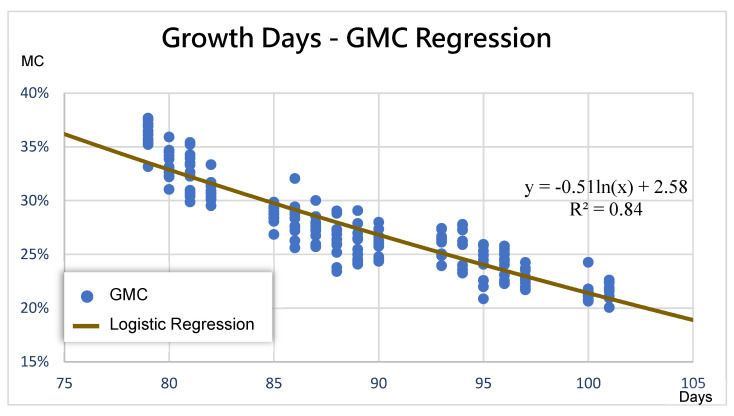
Data distribution and logistic regression of the growth days and GMC.

**Figure 18 sensors-21-05875-f018:**
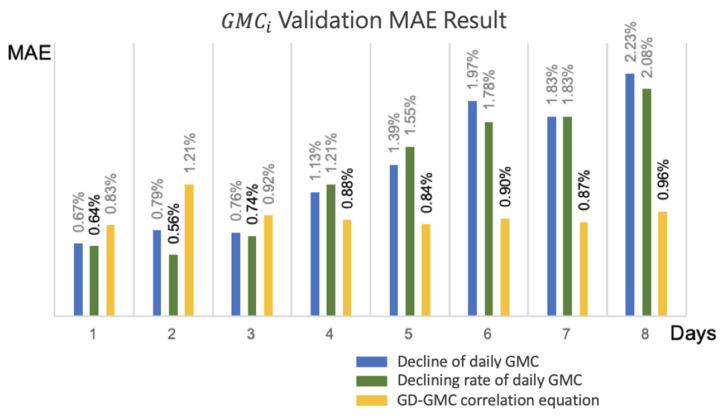
Multi-day validation data for $GMC_{i}$
*MAE* evaluation.

**Figure 19 sensors-21-05875-f019:**
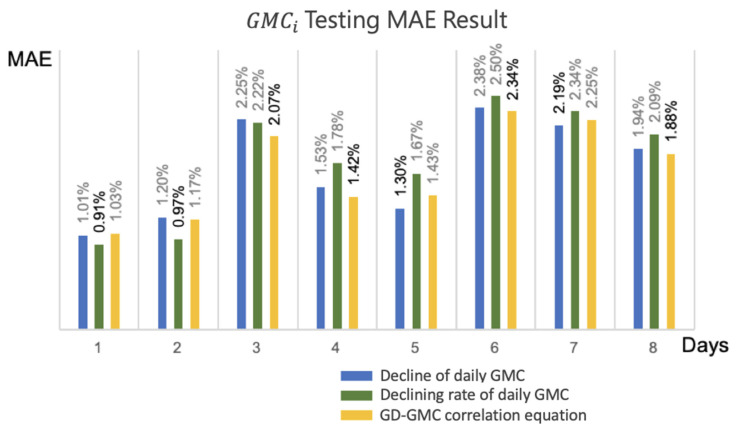
Multi-day testing data for $GMC_{i}$
*MAE* evaluation.

**Figure 20 sensors-21-05875-f020:**
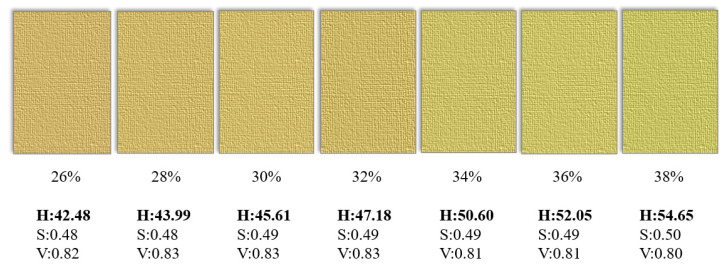
The schematic diagram of a panicle color board in term of HSV.

**Table 1 sensors-21-05875-t001:** Price reference of public stockholding programs by WuFeng Farmers’ Association.

HMC (%)	Purchase Price (TWD/600 g)	Note
>32%	ineligible	Volume weight > 560 (g/L) (non-weather abnormality, ineligible)
31–31.9%	997	Volume weight > 550 (g/L) (GIGR > 17%, ineligible)
30–30.9%	1025	
29–29.9%	1046	
28–28.9%	1067	
27–27.9%	1081	Volume weight > 540 (g/L) (GIGR > 17%, ineligible)
26–26.9%	1095	
<25.9%	1109	

**Table 2 sensors-21-05875-t002:** Smart phone and camera parameters.

Parameters	Values
Smart phone	Apple iPhone 8
Camera resolution	4032 × 3024 (12.1 M pixel)
ISO value	25
f/number	f/1.8
Shutter Speed	1/400 s
Still image aspect ratio	4:3
Spectral bands	3 (Red, Green, Blue)
Output formats	JPEG
Distance from lens to rice panicle	27.5 cm

**Table 3 sensors-21-05875-t003:** Result of the principal component analysis by color index.

Factor	Comp.1	Comp.2	Comp.3
*R*	−0.246	0.303	0.050
***G***	0.126	**0.392**	−0.084
***B***	−0.098	0.057	**−0.454**
***H*_1_**	**0.337**	0.118	0.008
***S*_1_**	−0.016	0.100	**0.464**
*V* _1_	−0.234	0.316	0.049
*H* _2_	0.337	0.118	0.008
*L* _2_	−0.211	0.234	−0.278
***S*_2_**	−0.187	**0.323**	0.176
***L****	0.025	**0.421**	−0.074
***a****	**−0.338**	−0.117	0.030
***b****	0.096	0.226	**0.384**
*Y*	−0.292	0.185	−0.141
*Cr*	−0.082	−0.251	−0.370
*Cb*	0.039	0.307	−0.326
*NDI*	0.334	0.086	−0.117
*GI*	0.335	0.087	−0.111
*RGRI*	−0.333	−0.086	0.124
**Component**	**Eigenvalues**		**Cumulative Proportion (%)**
Comp.1	8.049		44.71
Comp.2	5.466		75.08
Comp.3	4.381		99.42

**Table 4 sensors-21-05875-t004:** Performance comparison between RF, MLP, SVR, and MLR in different GMC intervals.

GMC Interval	Statistics Value	RF	MLP	SVR	MLR
All (*n* = 103)	*RMSE*	2.98	2.69	2.86	2.49
	*MAE*	2.04	1.77	1.79	1.71
	*MAPE*	6.28%	5.36%	5.24%	5.39%
Below 40 (%) (*n* = 86)	*RMSE*	2.15	1.82	1.74	1.90
	*MAE*	1.63	1.29	1.23	1.38
	*MAPE*	5.87%	4.71%	4.41%	5.04%
Below 32 (%) (*n* = 77)	*RMSE*	1.80	1.66	1.53	1.73
	*MAE*	1.41	1.20	1.08	1.31
	*MAPE*	5.37%	4.55%	4.10%	4.96%

**Table 5 sensors-21-05875-t005:** Comparison of GMC sensors with various operation principles.

Operation Principle	Portable Resistance	Resistance	Capacitance	Smartphone Image
Model	Kett Electric Laboratory FQ-527	Kett Electric Laboratory PQ-520	Kett Electric Laboratory PM-450	Apple Inc. iPhone 8
Recommended range	<20% *w.b.*	<20% *w.b.*	<20% *w.b.*	40–20% *w.b.*
Applicable scenarios	outdoor	indoor	indoor	outdoor
Weight	450 g	>9000 g	1300 g	148 g
Testing method	Destructive	Destructive	Destructive	Non-destructive
Sampling condition	Threshing	Threshing	Threshing	Directly shooting

**Table 6 sensors-21-05875-t006:** Extended testing result.

Testing Samples	Image-Based GMC Dataset	Samples of Different Growing Conditions	Samples of Different Varieties
Year	2019	2020	2020
Crop season	II	I	II
Variety	TNG71	TNG71	TN11
Characteristics	Middle-late maturity	Middle-late maturity	Early maturity
Assessment model	Image-GMC assessment model		
Testing *MAE*(% *w.b.*)	1.71	3.31	3.66

**Table 7 sensors-21-05875-t007:** Comparison of two on-site sampling methods.

	Image-Based GMC Dataset	Multi-Day GMC Dataset
Target model	Image-GMC assessment model	Multi-day GMC prediction model
Sampling scope	Single panicle	1 m × 1 m paddy
Feature	High variation GMC data	Homogeneous GMC data
Sampling frequency	2–3 days	1–2 days
Sampling (each time)	70–122	12
GMC distribution	60–13% *w.b.*	37.5–20.2% *w.b.*
Method	Drying	Drying
Drying spec	80 °C 7 days/each sample	80 °C 7 days/each sample
Labor costs	Three person/each day	Four person/each day

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
