# Peer review of "Assessment of Grain Harvest Moisture Content Using Machine Learning on Smartphone Images for Optimal Harvest Timing"

_sensors, 2021, doi:10.3390/s21175875_

Round 1
Reviewer 1 Report
L 19 give reference showing 80 C for 7 days provides an accurate measure of rice moisture
L 30 MAE needs to be defined
L 42 you need to state you are making a big assumption that photos that capture rice panicle maturity, mostly due to color, texture, etc are actually changed due to rice moisture content, and not due to other factors such as variety, fertility, availability of nutrients to the plant, time of year, plant moisture availability, etc. Your technique might work for one field with one variety and one set of conditions, but what about other varieties and other fields under different growing conditions?
L 68 Accurately measure GMC? Not sure that is proven yet.
L 96 hot wind? Perhaps hot air movement in an oven
L 75 OHD is it defined? You have a high number of 3 letter abbreviations, which distract the reader, because one has to go back and try to see what it means.
L 114 In terms of convenience, SP camera .. be convenient for your readers and spell out smart phone. You have an excessively large number of abbreviations.
L 180-184 are assumed, not needed.
L 214 what was the tebuconazole for? Why was etridiazole used?
L 218 seed density, is this the plant population when grown? State it in plants per a unit area.
Fig 3 caption define TARI
L 244 describe the background used for the images taken.
Table 2 give the distance from the camera lens to the object.
L 438 to 440 Is your grain moisture actually accurate to 0.01 % moisture? round numbers to nearest 0.1%. In reality accuracy is likely not even to the nearest 1%. Fig 11 and 12 round mc to 1.0%.
Fig 13 caption should state x value of Figs is grain moisture, in % wet basis.
Fig 16 and 17 How does your choice of Growth Days compare to the commonly used agronomic statistic of Growing Degree Days? Growing Degree Days accounts for sunlight and growing temperatures withing a certain growth range.
Table 5 caption, spell out two
Line 550 authors belabor a lot of discussion about the unsuitability of resistance type moisture meters, but most grain moisture meters used today are capacitance based, which is a total different principle of operation.
Your study should have objectives. Authors come close around line 176 but objectives were never concisely written. This is the problem for your conclusions section. The first paragraph in conclusions starts to conclude. The second paragraph is a summary. The third paragraph gives a review. So conclusions are not clearly or completely written.
This is actually a good study with a good proof of concept of using images to detect crop maturity and hence indirectly moisture content. Some of it is very wordy, and all of it uses too many abbreviations which detract from its readability for readers in a different discipline. Objectives of the study need to be clearly stated, then author’s conclusions need to address each objective and state how well each objective was met. Then pull out your most important conclusions and repeat them in the Abstract.
Author Response
"Please see the attachment."

Reviewer 2 Report
The present work shows a predictive model based on machine learning to determine rice grain moisture to predict the best harvest time.
The work is interesting and is relatively well supported by bibliographical references. However, it has some flaws that need to be clarified and improved, namely:
1. The abstract is excessively long, 400 words, more than double recommended for the journal.
2. The abstract should be revised to present the fundamental question, methods, results, and conclusions. If you do not follow this framework explicitly, it should effectively summarize and present the fundamental question.
3. A table with acronyms and acronyms must be presented, given the number of acronyms in the text, where some were not described, for example, "MAE".
4. The authors state that the images were collected with a smartphone and recorded in JPG format. This type of equipment is not the most suitable for this type of collection, as there are great disparities between the types of lenses and internal filters. In this case, why didn't they use a professional camera and record in RAW/TIFF format to avoid the loss of information introduced by the MPEG compression algorithm. With other equipment and image definition, the model may be quite skewed.
5. Formula (6) must be supported by a bibliographic reference, as this refers to the luminance formula, which allows obtaining Gray indices = 0.299 × R + 0.587 × G + 0.114 × B. It is recommended to refer to the page which supports Open-CV https://docs.opencv.org/3.4/de/d25/imgproc_color_conversions.html
Round 2
Reviewer 2 Report
Dear authors,
Thanks for the answer. In general, the main points were answered.
In answer to point 4: "The study had used mobile app Adobe Lightroom (v4.3.2) to collect RAW format images, and found that most mobile camera applications do not support RAW format... ", the authors can explain with more detail, that the use of JPEG algorithm doesn't lose quality nor make bias in the results.
Good luck.
